# Characteristics and Application of Eugenol in the Production of Epoxy and Thermosetting Resin Composites: A Review

**DOI:** 10.3390/ma15144824

**Published:** 2022-07-11

**Authors:** Danuta Matykiewicz, Katarzyna Skórczewska

**Affiliations:** 1Faculty of Mechanical Engineering, Institute of Materials Technology, Poznan University of Technology, Piotrowo 3, 61-138 Poznan, Poland; 2Faculty of Chemical Technology and Engineering, Bydgoszcz University of Science and Technology, Seminaryjna 3, 85-326 Bydgoszcz, Poland; katarzyna.skorczewska@pbs.edu.pl

**Keywords:** eugenol, bio-based, epoxy resin, thermoset resin composites, thermomechanical properties

## Abstract

The review article presents an analysis of the properties of epoxy and thermosetting resin composites containing eugenol derivatives. Moreover, eugenol properties were characterized using thermogravimeters (TGA) and Fourier-transform infrared spectroscopy (FTIR). The aim of this work was to determine the possibility of using eugenol derivatives in polymer composites based on thermoset resins, which can be used as eco-friendly high-performance materials. Eugenol has been successfully used in the production of epoxy composites as a component of coupling agents, epoxy monomers, flame retardants, curing agents, and modifiers. In addition, it reduced the negative impact of thermoset composites on the environment and, in some cases, enabled their biodegradation. Eugenol-based silane coupling agent improved the properties of natural filler epoxy composites. Moreover, eugenol flame retardant had a positive effect on the fire resistance of the epoxy resin. In turn, eugenol glycidyl ether (GE) was used as a diluent of epoxy ester resins during the vacuum infusion process of epoxy composites with the glass fiber. Eugenol-based epoxy resin was used to make composites with carbon fiber with enhanced thermomechanical properties. Likewise, resins such as bismaleimide resin, phthalonitrile resin, and palm oil-based resin have been used for the production of composites with eugenol derivatives.

## 1. Introduction 

In recent years, scientists have been looking for solutions to reduce the negative impact of polymeric materials on the natural environment. One of the widely described methods is the production of polymers from renewable sources and the introduction of various types of bio-additives into them [1,2]. Due to the fact that commercially used epoxy resins are made of petroleum materials, the development of a bio-resin for structural and engineering applications is very important [3,4]. Epoxy resins, also known as polyepoxides, belong to the group of thermoset materials containing a reactive oxirane group. Most often, epoxy resins are obtained by a direct method of polycondensation of epichlorohydrin with dihydroxy phenols or polyglycols in an alkaline environment. The resulting product is a bisphenol-A-based epoxy monomer, which, when combined with a curing agent (e.g., amines, acid anhydrides), forms an epoxy material [5]. Epoxy resin based on diglycidyl ether of bisphenol A (DGEBA) is most often brittle and not resistant to cracking; therefore, it is reinforced with various fillers and fibers in order to produce materials with higher strength [6]. The unique structure and properties of bio-renewable products that can be used in the production of epoxy resins with favorable thermomechanical and mechanical properties are still a challenge for scientists [7,8,9,10].

The use of natural materials as components for the production of epoxy resins has been widely described in the literature of recent years [11,12,13,14]. Cardanol from the cashew industry waste was used to obtain a resin system consisting of benzoxazine and epoxy prepolymers [15]. On the other hand, a cardanol derivative containing benzoxazine with boron-doped graphene was used to improve the fire resistance of the epoxy resin [16]. Moreover, epoxidized soybean oil was applied successfully in the production of epoxy vitrimers [17,18,19,20]. Nicolas et al. described the method of obtaining epoxy vitrimers from waste sunflower oil [21]. Xu et al. used renewable tung oil subjected to a methyl esterification reaction, a Diels–Alder reaction, and an epoxidation process to obtain a TO-based triglycidyl ester [22]. Bio-based tannic acid epoxy produced by modification of tannic acid was described by Borah et al. [23]. Likewise, vanillin-based epoxy oligomers are an important group of materials for the synthesis of biological thermosetting epoxy resins [24,25,26,27]. Dai et al. describe the use of genistein, which contains an isoflavone structure, as the basis for the production of a sustainable epoxy resin [28]. Another widely described renewable source for the production of bio epoxy resin is lignin [29,30,31]. 

Eugenol belongs to the group of organic chemical compounds of terpenes; it is a volatile phenolic component of clove oil obtained from Eugenia [32]. Due to its properties, it is popularly used in cosmetics, but in recent years, it has also been used in the production of biopolymers. Santiago et al. presented a method of producing a renewable epoxy resin from a trifunctional epoxy eugenol monomer [33]; Wan et al. prepared high rigid and low flammable epoxy polymer based on eugenol [34]; Jiang et al. synthesized thermosetting epoxy resins derived from eugenol and vanillin [35]; Chen et al. prepared epoxy compounds from the esterification of eugenol and performed a self-curing reaction of thermosetting epoxy resins [36]; Liu et al. described eugenol-based epoxy materials obtained by reaction with succinic anhydride and zinc-containing catalysts [37]; and Kumar et al. presented an overview of the synthesis of eugenol-based resins and their application as coating materials [38]. Importantly, the eugenol-derived epoxy materials exhibited favorable properties such as improved adhesion, corrosion protection, and a rigid structure [39,40,41,42].

The recent literature describes various methods of using eugenol for the synthesis of polymeric materials [43,44,45,46,47,48], including thermoset resins. However, at present, a limited number of articles are available describing the preparation methods and features of thermosetting composites that were made by incorporating eugenol as a component or modifier. The prospects of applying eugenol as a potential raw chemical resource for producing resins and other agents or additives are very broad. First of all, it should be noted that eugenol is a substance of low toxicity, and the methods of its preparation are simple. A common method was to isolate it from clove oil by steam distillation, extraction with methylene chloride, and finally distillation of the solvent [49]. The main component of clove oil is eugenol, and its concentration ranges from 70% to 96%. It is also found in cinnamon leaves, bay leaves, and nutmeg but in lower amounts [50]. Synthetically, it is obtained by reacting guaiacol with allyl chloride [51]. Simple synthesis methods also contribute to its low price. Eugenol (C_10_H_12_O_2_) is classified as an aromatic substance that belongs to the phenol group. Its structure includes the following functional groups: allyl (-CH_2_-CH=CH_2_), methoxy (−OCH_3_), and phenol (OH) [52]. As such, eugenol is susceptible to many chemical and biochemical reactions. Eugenol has many uses, including in dentistry, perfumery, the food industry, and in recent years, as a bio component of polymer materials.

Therefore, the aim of this review is to characterize epoxy and other thermosetting resin composites based on eugenol derivatives and their properties, which, according to the author’s best knowledge, have not been reported so far because work on completely renewable resins obtained from eugenol is in progress.

## 2. Chemical Characterization of Eugenol Derived Resins and Auxiliary Components for Composites

Obtaining polymers from eugenol has attracted the interest of both chemists and materials scientists [53,54]. The presence of a hydroxyl group and an allyl group in the eugenol molecule makes it a compound capable, i.a., of etherification, esterification, or thiolene reaction [55]. Eugenol may be used to prepare reactive polymer monomers, hardeners, modifiers, and auxiliaries that successfully replace synthetic substances of petrochemical origin. Due to the decreasing oil resources, the greenhouse effect, and environmental pollution, there is a growing interest in the use of bio-based resources on a laboratory and industrial scale [56]. The use of natural phenolic chemicals such as eugenol and its derivatives could create an opportunity for the production of polymer materials as part of a sustainable circular economy. 

### 2.1. Characteristics of Eugenol

Eugenol containing 100% clove oil (Chema Elektromet, Rzeszów, Poland) was subjected to Fourier-transform infrared spectroscopy (FTIR) analysis using a spectrophotometer Alpha (Bruker, Billerica, MA, USA) with ATR accessory in the range from 400 to 1800 cm^−1^ and thermal gravimetric analysis (TGA) using TG 209 F3 Netzsch (Selb, Germany) in an atmosphere of nitrogen and in the temperature range 30–900 °C with a heating rate of 10 °C/min. The following data were appointed: the temperature at which the weight loss was 5% (T5%) and 10% (T10%) and the maximum thermal degradation temperature from the derivative thermogravimetric (DTG) plot. Figure 1 shows the characteristic chemical structure of eugenol with the preferred IUPAC name 2-Methoxy-4-(prop-2-en-1-yl)phenol. Eugenol (C_10_H_12_O_2_) is an oily liquid with a characteristic intense clove odor, a molecular weight of 164.2 g/mol, and is soluble in methanol, acetic acid, and diethyl ether. 

Figure 2 presents the FTIR curve for eugenol with characteristic peaks: in the range of 720–1250 cm^−1^ attributed to the *v*C=C bond, the sharp peaks visible at 1600 and 1500 cm^−1^ correspond to the stretching of C=C of the aromatic group, peaks at a wavelength in the range 2700–2900 cm^−1^ can be assigned from CH_3_ groups and peak at 3500 cm^−1^ comes from hydroxyl groups [57,58].

The TGA and DTG curve of eugenol is presented in Figure 3. The temperature at which the weight loss was 5% and 10% was observed at 134 °C (T5%) and 147 °C (T10%), respectively, and can be attributed to the vaporization of eugenol. One sharp peak at 205 °C was observed on the DTG curve, indicating the degradation of eugenol with a maximum degradation rate of 27.4%/min. Eugenol is an organic substance; therefore, no residual mass was detected.

### 2.2. Characterization of Eugenol Derived Compounds

Eugenol-based epoxy resins have some disadvantages: lower glass transition temperature (Tg) than bisphenol A diglycidyl ether (DGEBA), lower viscosity, and higher hydrophilicity [37,59]. Therefore, they are modified and combined with other fillers or reagents, creating polymer composites with favorable properties. The main methods of making eugenol-based composites include mixing, cross-linking, and introducing reinforcement particles [38]. On the other hand, researchers found that eugenol-based resins compared to petroleum epoxy resin, have excellent anti-corrosion, antioxidant, tensile strength, adhesion, and thermal stability [60,61]. The list of eugenol derivatives used in the production of composites presented in the review is presented in Table 1. The description of the synthesis of these compounds can be found in the section describing the properties of the composites.

## 3. Properties of Composites Based on Eugenol Derived Resins and Additives and Coupling Agents

### 3.1. Epoxy Resin Composites

Epoxy resins, due to their simple processing and high compatibility with different types of fillers, can be used as the polymer matrix in different composite materials. The properties of cured epoxy resin depend on the chemical structure of the curing agent, the curing process conditions, and the influence and reactivity of modifiers. This is the reason why the selection of the right filler in order to obtain a composite of required chemical and thermal resistance, as well as advantageous mechanical properties, remains an important subject of scientific research. The increase in the application of polymer composites such as light construction materials with high durability in the automotive, aviation, shipping, and construction industries results in intensified applied and industrial research into new material and technological solutions regarding the processing of such materials. Cured epoxy material is characterized by high resistance to weather and chemical factors. At the same time, such materials are highly brittle, which requires mixing them with various fillers (such as fiber and fabric), which allows for obtaining construction materials of high stiffness and durability. In order to reduce the negative impact of epoxy materials on the environment, many studies concern the use of natural fibers as reinforcement in composites [73,74,75]. Another method of producing ecological epoxy composites is the use of bio-based curing agents, among others, from vegetable oils [3].

Environmentally friendly composites include polymers and fillers that are either biodegradable or of biological origin. Biocomposites can be grouped according to composites made of renewable raw materials; composites, the waste of which can be composted or recycled; and composites whose production process is sustainable [76]. It is most preferred that all components of the composite material are derived from renewable sources. Therefore, research into the use of eugenol as a bio component of epoxy resin composites for industrial applications is very important. Articles describing epoxy composites in which eugenol was used to modify the polymer matrix or filler and as a modifier or hardener were selected for the review.

Due to its high reactivity with other compounds, eugenol can be used to modify fillers or as a component for the synthesis of polymer monomers. The method of connecting silicone with bio-based derivatives to obtain silicone–epoxy networks was described by Li et al. [77]. Eugenol was exposed to the epoxidation process during glycidyletherification between a phenolic hydroxyl group and epichlorohydrin. The epoxidized eugenol (EPEU) was then hydrosilylated to form epoxy monomers with silicone linkages named SIEEP2, SIEEP4, and SIEPEP. The monomers thus obtained were hardened with diaminodiphenyl sulfone (DDS). The introduction of methylsiloxane and phenylsiloxane segments to the epoxy network improved the properties of composites, such as excellent dielectric permittivity and flame retardancy, with a Limiting Oxygen Index (LOI) of up to 31. 

A reference to the above research is the article by Aziz et al., in which he described the process of the synthesis of eugenol-based silane coupling agent (EBSCA) and applying it to modify cellulose nanocrystals (CNC) [62]. Eugenol-based silicone coupling agent was obtained by the hydrosilylation between triethoxysilane and epoxidized eugenol. Tiethoxysilane, toluene, and Karstedt catalyst were mixed under the nitrogen flow for 1 h to remove oxygen. When the hydrosilylation reaction ended, the Karstedt catalyst was removed by flash chromatography [63]. The modified CNC was then incorporated into a bio-based liquid epoxy resin (SIEEP4 and SIEEP2) at a concentration of 1%, 3%, and 5%, and then the composition was mixed with a triethylenetetramine hardener (TETA) to produce composites. The method of producing the composite consisted of CNC sonication in the matrix, mixing with a hardener, casting into steel molds, and hardening at an elevated temperature of 120 °C/1 h. SIEEP2 and SIEEP4 have different dimethylsiloxane segment lengths and were made of eugenol and functional silicone monomers [62]. The addition of a higher amount of modified CNCs increased the surface roughness of the composites. The authors also assessed the average adhesive strength for an epoxy composition containing eugenol-based silane coupling agent modified CNCs and triethylenetetramine as a curing agent. The maximum value was recorded for the 1% CNC modified composite assessed for modulus of elasticity and strength at maximum load when using steel plates. Composites with 1% CNC modified from EBSCA showed increased mechanical properties such as high tensile strength (2190 MPa) and modulus (16.00 MPa), as well as the storage modulus (1622 MPa) and degradability Eugenol epoxy silane coupling agent, with a long-chain structure of the benzene ring in the molecule that contributes significantly to improving the compatibility of cellulose nanocrystals with other epoxy matrices, at the same time increasing the application potential of these materials. 

On the other hand, Zheng et al. developed nano-SiO_2_ epoxy composites treated with eugenol epoxy silane (EUPCP), which was obtained via the hydrosilylation between triethoxysilane and 3-(4-allyl-2-methoxyphenoxy) propylene 1,2-oxide EPEU [63]. The modified nano-silica was incorporated into a liquid epoxy resin (DEGBA—diglycidyl ether of bisphenol A) by ultrasonication dispersion and stirring. Then, the mixture prepared in this way was combined with the isophorone diamine hardener (IPDA) and was poured into molds and cured at elevated temperatures (80 °C/2 h and 150 °C/2 h). Composites with a content of 1, 2, and 4 wt.% unmodified and modified SiO_2_ were produced. Among the tested materials, the best properties, such as higher glassy storage modulus and the glass transition temperature as well as thermal stability, were demonstrated by the composite containing 4 wt.% modified SiO_2_. Moreover, epoxy composites with 4 wt.% modified SiO_2_ exhibit flexural strength, impact strength, and KIC increased by 7.2%, 17.4%, and 6.3% comparison to epoxy composites with 4.0 wt.% unmodified nano-SiO_2_. The authors confirmed that the modified nano-SiO_2_ particles, showing good dispersion in the epoxy matrix, significantly contributed to the improvement of the thermal and mechanical properties of composites.

Vaithilingam et al. described the method of obtaining the eugenol–benzoxazine monomer (EBUz) based on thiourea and its use in the production of epoxy composites with amine functional carbon from cashew nut shells (f-CSC) [64]. Eugenol was mixed with paraformaldehyde and refluxed at 80 °C for 1 h. Next, thiourea was introduced to composition, and the reaction was conducted at 100 °C for 12 h to obtain EBUz. The scheme of preparation of these composites is shown in Figure 4. For a composition containing a 1:1 ratio of EUBz and diglycidyl ether of bisphenol-A (EP), and the calculated amount of the polyamide–imidazoline hardener was introduced to f-CSC, then the mixture was poured onto a glass plate and hardened (according to the schedule 40 °C, 60 °C, 80 °C, 100 °C, 120 °C, 160 °C and 180 °C for 1 h at each temperature) to obtain thin films. The produced composites based on thiourea-based eugenol-EUBz monomer and epoxy resin (EP) containing 1, 3, and 5 wt.% f-CSC were characterized by improved thermal stability, greater char yield, and higher Tg values. The authors showed that the glass transition temperature, the initial degradation temperature, and the char yield were the highest for the samples with the highest filler content and were, respectively, 202.9 °C, 332.8 °C, and 45.28%. In addition, composites indicated good hydrophobic and anti-corrosion futures. The authors showed by testing the contact angle that the increase in f-CSC content in polybenzoxazine EP composites (for which there was an increase in contact angle values from 95.1 to 110.8) contributes to the improvement of water and moisture resistance properties and makes it possible to use these materials as anti-corrosion coatings.

On the other hand, Selvaraj et al. used eugenol to prepare the hetero structured benzoxazine monomer (HSBBz) [65]. HSBBz was obtained in the reaction of cardanol, eugenol, and paraformaldehyde, which were mixed and taken in a two-neck round-bottomed flask and refluxed at 65 °C for 1 h. Then, diphenyldiaminomethane (DDM) was added, and compositions were stirred at 80 °C; next, after 15 min, methanol was added, and the stirring was continued at 60 °C for 72 h. Finally, the reaction mixture was poured into sodium hydroxide, and the benzoxazine monomer was separated using chloroform. Moreover, the organic layer was dried over sodium sulfate and filtered. Then benzoxazine and epoxy resin (EP) were then combined in a ratio (1:1) to form biocomposites with functionalized cow dung carbon (*f*-CDC). The composites contained 1, 3, and 5 wt.% of f-CDC and were cured with polyamide imidazoline at elevated temperatures (according to the scheme at 40 °C/1 h, 60 °C/1 h, 80 °C/ 1 h, 100 °C/1 h, 120 °C/1 h, 160 °C/1 h and 180 °C/1 h). Composites indicated high thermal stability and high char yield with an increasing amount of f-CDC in the polymer matrix. Moreover, the glass transition temperature value of composites was higher than that of the reference sample, and for the sample with 5 wt.% of f-CDC, it was 199.9 °C. In addition, the authors showed that the presence of f-CDC in the HSBC-EP matrix may limit the growth of bacterial activity and cause cell loss in bacteria. As in the case of composites with f-CSC, the addition of small amounts of f-CDC especially enhanced the anti-corrosion properties of the HSBBz-EP matrix. The research confirmed that these biocomposites could be used as an efficient coating material and, at the same time, create the possibility of utilizing carbon materials obtained from biomass. 

The self-repairing properties of internal damage in composite materials is a highly requested feature. Therefore, Li et al. developed an organosilicon-modified epoxy monomer (EETS) obtained from eugenol and 1,1,3,3-tetramethyldisiloxane and cured by 4-aminophenyl disulfide (APDS) to prepare dynamic epoxy resins with a disulfide bond (EETS/APDS) [66]. This material was obtained by melting polymerization of the epoxy group and amidogen with the disulfide bond. The composition was heated to 80 °C and stirred for 10 min, then poured into heated molds, degassed, and cured according to the scheme at 110 °C, 140 °C, 170 °C, and 200 °C for 2 h, appropriately. The EETS/APDS composites were characterized by a low glass transition temperature (53 °C), tensile strength (39 MPa), and high elongation at break (22%) related to commercial epoxy materials. In addition, bio-based epoxy resins indicated thermal stability to 287 °C, which is close to the thermoset of diglycidyl ether of bisphenol, for which the initial decomposition temperature was 304 °C. Importantly, the tested resins showed a high self-healing property, reprocessing, and degradation ability. The authors proved that the deformed samples could be straightened by heating them to 80 °C for 10 min. This phenomenon is attributed to the dynamic exchange of disulfide bonds. In addition, the power of the EETS/APDS composites may be reshaped by the hot press process at 200 °C under 20 MPa. Moreover, full degradation of this material can be performed by ultrasonic vibration in a solution of m-CPBA and DMF at 30 °C for 5 h. These materials are an advantageous alternative to petrochemical resins and their composites.

Due to the flammability of the epoxy resin and a large amount of smoke emitted, the development of a bio-based flame retardant is the subject of many scientific papers. Reactive phosphate flame retardant derived from isoeugenol was developed by Pourchet et al. [68]. Environmentally friendly flame retardant such as diepoxy-isoeugenol phenyl phosphate (DEpiEPP) was prepared. This unique compound combines both the bio origin isoeugenol (iEu) catalytically extracted from the lignin and a phosphate group, providing flame retardancy. DEpiEPP was synthesized in two steps: condensation of iEu and phenyldichlorophosphate (PPDC) in toluene with triethylamine (TEA) as acid acceptor and epoxidation by oxone. Then, it was introduced into diglycidyl ether of bisphenol A resin (DGEBA) and into the bio-glycidyl ether of epoxyisougenol (GEEpiE) and cured with camphor anhydride (CA) for 2 h at 150 °C. Thermal analysis proved that DEpiEPP is a good flame retardant and provokes a significant charring of both epoxy thermosets (DGEBA-DEpiEPP and GEEpiE-DEpiEPP). In the case of DGEBA/CA and GEEpiE/CA materials, along with the increase in phosphorus content in the flame retardant, the amount of char content also increased. The glass transition temperature determined via DSC analysis of the cured composites decreased with increases in the amount of DEpiEPP monomer from 155 °C to 71 °C in the DGEBA matrix and from 162 °C to 71 °C in GEEpiE. Importantly, for all tested composites, no additional exothermic peak was observed on the DSC curves after the hardening process, which proves that the cross-linking reaction was completed. Analysis of the mechanical properties showed that the DGEBA-CA-DEpiEPP epoxy polymer exhibits ductile behavior with a yield point of about 85 MPa and 2% strain, followed by highly nonlinear behavior suitable for plasticity. While the GEEpiE -CA-DEpiEPP epoxy system exhibits elastic–brittle properties with sudden failure after exceeding the yield point. The Tg value of the composites containing 2% phosphorus was 105 °C for the DGEBA matrix and 129 °C for the GEEpiE matrix, and the modulus of elasticity was 5.1 and 4.8 GPa, respectively. Such features are suitable for the production of composites with favorable thermomechanical properties. Additionally, Yu et al. developed a eugenol-based flame retardant containing phosphorus (P) and silicon (Si) (EGN-Si/P) groups for bisphenol A epoxy resin [69]. The synthesis scheme of EGN-Si/P is shown in Figure 5. Flame retardancy was obtained by a two-step reaction from eugenol. First, EGN-P was synthesized by the Williamson etherification reaction from a solution of eugenol and triethylamine in tetrahydrofuran i diphenylphosphinyl chloride. Then, the platinum catalyst was dissolved in methylbenzene and heated then silane was added. 

The method of preparation of the composites included: mixing EGN-Si/P with curing agent 4,4-diaminodiphenyl methane (DDM) and resin. Then composition was heated to 95 °C for 3 min, degassed, and next poured into the mold and cured at 100 °C/2 h, 130 °C/2 h, and 160 °C/2 h. The modified resin contained 0.5% Si and 0.37% P% (0.5EGN-Si/P); 1% Si and 0.74% P (1EGN-Si/P); and 2% Si and 1.48% P (2EGN-Si/P). The Tg of the modified resin lowers as the addition of EGN-Si/P decreases from 175 °C to 148 °C. The tensile strength of the modified samples (0.5EGN-Si/P and 1EGN-Si/P) was similar to the reference DGEBA resin and at about 60 MPa. However, for the sample 0.5EGN-Si/P, a significant increase in impact strength was observed, which amounted to 23.28 kJ/m^2^ and was 45.6% higher than for unmodified resin. The authors showed that EGN-Si/P has good flame retardancy. Moreover, the introduction only 0.5 wt.% Si/P addition to resin allows for achieving the V-0 level in the UL-94 test and the 28% limiting oxygen index (LOI) value. Importantly, the EGN-Si/P efficiency in suppressing the release of heat and smoke is higher than for the unmodified resin. The eugenol-based P and Si synergistic flame retardants make it possible to use environmentally friendly flame retardants for polymer composites. 

Epoxy resins, after the curing process, constitute a brittle, non-melting material with good chemical resistance. Nevertheless, in order to obtain the composites with high mechanical properties, they were combined with reinforcing fibers, usually manufacturing laminates. Yue et al. used biobased diphenolate diglycidyl ethers (DGEDP) and eugenol monoglycidyl ether (GE) as a diluent for the production of twelve-layer epoxy composites reinforced with E-glass fiber mats by vacuum infusion methods [70]. The method of obtaining glycidyl eugenol was described as a one-step glycidylation reaction with excess epichlorohydrin under alkaline conditions. The composition was 15 wt.% GE and 85 wt.% of either DGEDP-ethyl or pentyl esters and was cured with isophorone diamine (IPDA). The storage modulus of the DGEDP-ethyl/GE composite at 25 °C (13.7 GPa) was 11% higher than that of the commercial epoxy matrix composites Hexion (12.3 GPa). Figure 6 shows the thermomechanical curves of the tested composites. The glass transition temperature values of DEGDP-ethyl/GE and DGEGDP-pentyl/GE epoxy/glass materials were lower than those of commercial resins and amounted to 111 °C and 96 °C, respectively. However, it can be assumed that such temperatures are suitable for many industrial applications that do not require high operating temperatures. Moreover, the DGEDP-ethyl and DGEDP-pentyl composites showed a better flexural modulus (15.47 and 15.78 GPa) than the reference sample (14.76 GPa). These studies showed that eugenol glycidyl ether (GE) could be successfully used as a reactive diluent of DGEDP epoxy ester resins, allowing them to be processed by vacuum infusion for the production of fiber-reinforced composites.

Reddy et al. used eugenol-based epoxy resin to prepare epoxy composites with carbon fiber [67]. Eugenol, triethylamine, and dichloromethane (DCM) were added to a flask with a nitrogen inlet, stirrer, and thermocouple. Then 1,3,5-Benzenetricarbonyl trichloride in DCM solution was dropped at 3–5 °C for 1 h with quick stirring and then mixed at 25 °C for 12 h [78]. The white salt thus obtained was mixed with DCM and meta-chloroperoxybenzoic acid at 0 °C. The composition was stirred for 12 h at 40 °C; then, the insoluble parts were filtered and extracted with 10 wt% of Na_2_SO_3_. The obtained layer was dried over magnesium. Epoxy resin: (tris (2-methoxy-4-(oxiran-2-ylmethyl) phenyl) benzene-1,3,5-tricarboxylate) was mixed with 4-dimethylaminopyridine (DMAP) and dioxane, and then carbon fibers were soaked in this solution and dried at 70 °C. The composites consisted of six layers of carbon fabric and were produced by pressing at an elevated temperature in the range of 160–200 °C. The composites were thermally stable and indicated a decomposition temperature of 408 °C. In turn, DMA analysis showed that the composite modulus was at 10 GPa up to a temperature of 250 °C. This is a value 1000 times higher than that of pure eugenol-based epoxy. Moreover, the authors proved that after the degradation of the epoxy matrix by the aminolysis method, the carbon fiber could be recovered. SEM analysis showed that the recycled carbon fiber looks similar to the original one (Figure 7). The photographs show virgin carbon fibers (Figure 7c,e) and recycled carbon fibers (Figure 7d,f). It is worth noting that the recycled carbon fiber surface was clean and undamaged. Moreover, the mechanical properties of the carbon fiber monofilament did not deteriorate as a result of the recycling process. This makes it possible to safely recover carbon fibers from end-of-life composites.

Mattar et al. fabricated epoxy composites reinforced with recycled carbon fibers based on resorcinol diglycidyl ether (RE) cured with hexamethylenediamine (HMDA), Diamine-limonene (DA-LIM), and diamine-allyl-eugenol (DA-AE), respectively [72]. Diamine-allyl-eugenol was synthesized from eugenol by the Williamson reaction [79] and then thiol−ene radical reaction [71]. Allyl eugenol and cysteamine hydrochloride were dissolved in ethanol in a round-bottom flask, then 2,2′-azobis-isobutyronitrile was added, and the mixture was degassed under an argon flow. In the next stage, the mixture was heated in an oil bath at 75 °C/24 h. The DA-AE thus obtained was dried on anhydrous MgSO_4_, filtered, and evaporated to recover the pure product as a pale yellow oil. Composites were produced using the vacuum-assisted resin infusion method with a fiber volume content of 27–32% and cured for 2 h at 60 °C and 100 °C for 24 h at autoclave. The compositions RE/DA-AE and RE/DA-LIM showed suitable viscosity values (386 MPa·s at 40 °C and 1720 MPa·s, respectively) for the resin infusion method. Importantly, the TGA analysis confirmed that the introduction of carbon fiber for all tested types of resins improved their thermal stability by 30–35 °C. Moreover, these fully biological resins had higher gelation temperatures than DGEBA/HMDA and RE/HMDA and. The RE/DA-AE composites showed high tensile strength σ_Y_ (134 MPa) and bending strength σ_F_ (264 MPa) compared to the composites based on the commercial DGEBA/HMDA resin. The authors proved that the addition of recycled carbon fibres improves the thermal resistance of the tested composites; this is especially important for the application of bio resins, which usually have lower thermal resistance. Moreover, the paper presents an effective method of recycling carbon fibre from composites through the use of aminolysis.

### 3.2. Thermosetting Resins Composites 

The group of thermoset resins includes mainly polyester, epoxies, phenolic, melamine, and polyurethanes, which form three-dimensional networks as a result of cross-linking reactions under the influence of heat, cure agents, or UV radiation [80]. From among thermosetting composites, those whose polymer matrices were modified with eugenol derivatives were selected for review. Zhang et al. synthesized eugenol-functionalized polyhedral oligomeric silsesquioxane as a modifier for bismaleimide resin (BD) [81]. 

The produced composites contained 4 wt.% EG-POSS and were characterized by a homogeneous structure. Importantly, the dielectric constant and loss factor at 1 MHz decreased to 2.88 and 0.010 from 3.33 and 0.013. In addition, the hybrid material (BDEP) showed high thermal stability with a glass transition temperature of 290 °C and enhanced impact strength and flexural strength. Therefore, BDEP composites with silica fiber (SF) indicate high wave transmission efficiency (94%), flexural strength (310 MPa), flexural modulus (15.7 GPa), and interlinear shear strength (35 MPa) compared to BD/SF composites. The authors proved that eugenol-functionalized polyhedral oligomeric silsesquioxane could be successfully used to improve the properties and structure of fiber-reinforced composites. Likewise, Shibata et al. used eugenol (EG) for the synthesis of bieugenol (BEG) and eugenol novolac (EGN) and then as components of bio-based thermosetting bismaleimide resins [82]. BEG was obtained by the oxidative coupling reaction of EG, and EGN was obtained via the addition-condensation reaction of EG with formaldehyde. Next, EG, BEG, and EGN were used to make composites with 4,4′ -bismaleimidediphenylmethane (BMI) with eugenol/maleimide ratios of 1/1, 1/2, and 1/3. Cured resins were prepared by the compression-molded method according to the scheme at 200 °C/ 8 MPa during 1 h, 210 °C/8 MPa during 2 h, and 250 °C/8 MPa during 6 h. SEM analysis confirmed that all cured resin materials showed no phase separation and that EG, BEG, and EGN had reacted with BMI. It was noted that Tg and 5% weight loss temperature (T_5_) values for all the tested materials increased with increasing BMI addition. The eugenol (EG) based system showed the highest Tg (377 °C) and T_5_ (475 °C) values. It should be emphasized that the Tg and T_5_ values for eugenol-based systems and eugenol novolac were higher than the reported value for a commercial thermosetting bismaleimide resin. Among the tested systems, the most favorable properties are flexural strength (84.5 MPa) and modulus (2.75 GPa) and showed a composite based on (EG) with eugenol/maleimide unit ratios of 1/2. The authors presented an effective method of using eugenol and eugenol novolac in thermoset materials.

Ning et al. used eugenol (EG) and hydroxymethylated eugenol (MEG) as a bio-component for the synthesis of phthalonitrile resin (PN) [83]. Bio-based resin (MEG-PN) has been used in the production of fourteen-layer carbon fabric reinforced laminates. The production method consisted of impregnating the carbon fiber with MEG-PN resin at 80 °C, then hydraulic pressing at 200 °C and curing at 230 °C for 2 h at 4 MPa and 280 °C for 4 h at 6 MPa. The main advantages of the MEG-PN resin are a lower curing temperature (281 °C) without the need to use a hardener and good thermal stability compared to the PN resin. In addition, MEG-PN showed adequate viscosity at room temperature for the composite production technology, and the initial temperature of the process was also low and amounted to 77 °C. Moreover, carbon fiber composites based on this bio matrix showed favorable thermomechanical properties such as high glass transition temperature (397 °C) and high flexural strength (776 MPa) and modulus (119 GPa). The study confirms that MEG-PN resin can be successfully used to produce composites with favorable thermomechanical properties. 

In turn, Wu et al. used eugenol (EM) to prepare a bio-cross-linker and then to produce thermosetting composites based on palm oil (PO) reinforced with bamboo fiber [84]. A vinyl PO-based monoglyceride (POFA-EA) was obtained from palm oil by a transesterification process. In addition to eugenol–methacrylate (EM), gallate–methacrylate (GM), and tannic acid–methacrylate (TM) were used as cross-linkers. The bio-based resin was mixed with appropriate cross-linkers and with an initiator methyl ethyl ketone peroxide (MEKP) and an accelerant cobalt naphthenate. The composites were made by compression molding from unidirectionally oriented fiber mats and POFA-EA resin, with fiber to resin ratio set at 1:1 by weight. The authors assumed that one EM molecule might provide one cross-link point for the system, while both GM and TM particles with multiple C=C bonds could provide more cross-link points to the resins. The thermomechanical properties of the composites were similar to those of commercial materials. The Tg of the bamboo fiber composites with EM, GM, and TM as cross-linkers were 105.9, 165.4, and 171.5 °C, respectively. The tensile and flexural strength was, respectively, about 30 MPa and 32 MPa for composites with EM, 52 MPa and 80 MPa for composites with GM, and 62 MPa and 100 MPa for composites with TM. The work shows the way of using green materials to produce composites with favorable properties. The authors also predict that biocomposites completely degrade as PO matrices can be degraded under mild alkaline conditions. 

## 4. Discussion

Eugenol, due to its reactivity and easy availability, is an attractive component of thermosetting resin composites. Its low toxicity and the method of obtaining from renewable plant sources contribute to the growing interest in this modifier in order to produce environmentally friendly materials according to the assumptions of the sustainable development economy. Many studies show the use of eugenol to obtain bio epoxy resins, but there is no review of composites based on such a matrix. Eugenol-based resins have found application as coating materials with favorable anti-corrosion and adhesive properties. However, due to the fact that this resin is not yet produced on an industrial scale, it is important to research its application potential in composite materials. It can be estimated that about 10–15% of the total volume of plastic production consists of thermosetting resins, which is an important forecast that production on the world market will increase by 40% in the coming decade [85]. The increase in the use of thermosetting materials affects the amount of waste generated at the end of the life cycle. Therefore, one of the key assumptions of green chemistry is to increase the use of natural materials in the composite production process and thus minimize the amount of waste [86]. Figure 8 shows the possibilities offered by the use of eugenol derivatives to create composite materials. The diagram shows the main paths of eugenol application as a component used in the production of biomaterials, such as coupling agents, curing agents, diluents, flame retardants, and monomers. Composite is a material composed of a matrix and a filler, it is desirable that both of these components should be natural, biodegradable or their production process uses the available biomass and renewable sources. Due to the fact that eugenol can be used as a component of a polymer matrix or as an auxiliary agent in the process of obtaining composite materials, Table 2 lists the types of epoxy bio-systems analyzed in this review. The main methods of producing these bio-based epoxy composites were blending and casting, and for fiber-reinforced composites, vacuum infusion and pressing. The eugenol-based coupling agent improved the adhesion between the components of epoxy composites as well as their mechanical and thermomechanical properties. On the other hand, epoxy composites based on eugenol–benzoxazine monomer with a natural filler showed increased thermal stability and glass transition temperature. In addition, the eugenol-based flame retardant has proven effective in improving fire resistance. For fiber-reinforced composites, eugenol derivatives have found use as a diluent, hardening agent, and monomer component. The combination of the bio-based epoxy matrix with carbon fibers made it possible to create lightweight materials with favorable mechanical and processing properties, which can be completely degraded after the end of the product life cycle. In turn, for bismaleimide resins, eugenol was used as a bio component and as a modifier in the form of eugenol-functionalized polyhedral oligomeric silsesquioxane. In both cases, a significant improvement in mechanical properties such as flexural strength and modulus and glass transition temperature was achieved. Eugenol derivatives were also used to prepare phthalonitrile resin and then to produce carbon fabric reinforced composites. The materials obtained in this way were characterized by increased flexural strength and modulus as well as the glass transition temperature. In addition, eugenol as a bio-crosslinker was used in the preparation of the thermosetting composites based on palm oil reinforced with bamboo fiber. 

Such a wide area of application of eugenol and the above described favorable properties of ecological composites give the possibility of further work on this material and its popularization not only on a laboratory scale but also on an industrial scale.

## 5. Conclusions

Due to its reactivity, non-toxicity, and easy availability, eugenol has in recent years become the subject of many scientific works in the field of materials engineering. In particular, it is used in the production of composite materials based on thermoset resins as a component of coupling agents, epoxy monomers, flame retardants, curing agents, and modifiers. The following eugenol derivatives were used as an auxiliary in the processing of epoxy resin:Silane coupling agent based on eugenol, used for natural fillers such as cellulose nanocrystals, and nano-SiO_2_, leading to the improvement of thermo-mechanical properties of epoxy composites with their addition;Flame retardants such as diepoxy isoeugenol phenyl phosphate and a eugenol-based flame retardant containing phosphorus and silicon provide high flame retardancy to epoxy materials;Eugenol glycidyl ether is used as an effective diluent for epoxy-ester resins during vacuum infusion used in the production of epoxy composites with glass fiber;Diamine-allyl-eugenol is a hardener for epoxy resins based on resorcinol, providing higher gelling temperatures for this system and increasing the use of this material as a matrix for carbon fiber reinforced materials.

Moreover, eugenol as a component was used for the synthesis of the following monomers:Eugenol–benzoxazine monomer as a matrix for composites with carbon from cashew nut shells;Hetero structured benzoxazine monomer as a matrix for composites with functionalized cow dung carbon;Organosilicon-modified epoxy monomer with self-repairing properties;Epoxy resin (tris (2-methoxy-4-(oxiran-2-ylmethyl) phenyl) benzene-1,3,5-tricarboxylate) as a matrix for composites with carbon fibre.

Furthermore, for the production of composites with eugenol derivatives, the following resins were used: bismaleimide resin, phthalonitrile resin, and palm oil-based resin. The development of methods of using eugenol derivatives in the creation of composite materials plays a significant role in the development of green chemistry and limiting the negative impact of winder materials on the environment. Knowledge of the beneficial properties of eugenol and its processing potential should be disseminated in order to increase the production of polymeric materials from sustainable sources. Both the eugenol-based auxiliaries and monomers have shown favorable processing properties and can be successfully used in the preparation of composites.

## Figures and Tables

**Figure 1 materials-15-04824-f001:**
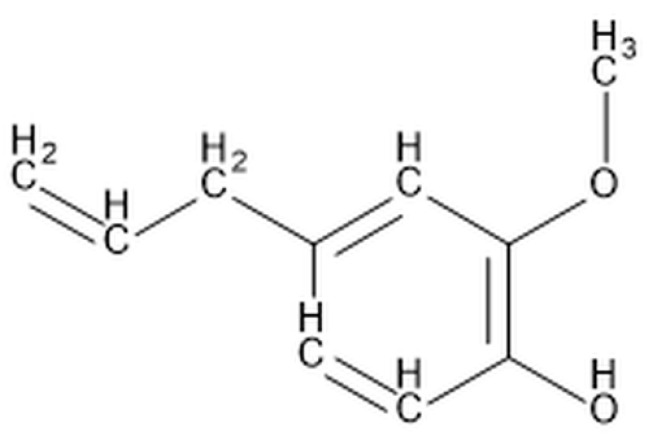
Chemical structure of eugenol.

**Figure 2 materials-15-04824-f002:**
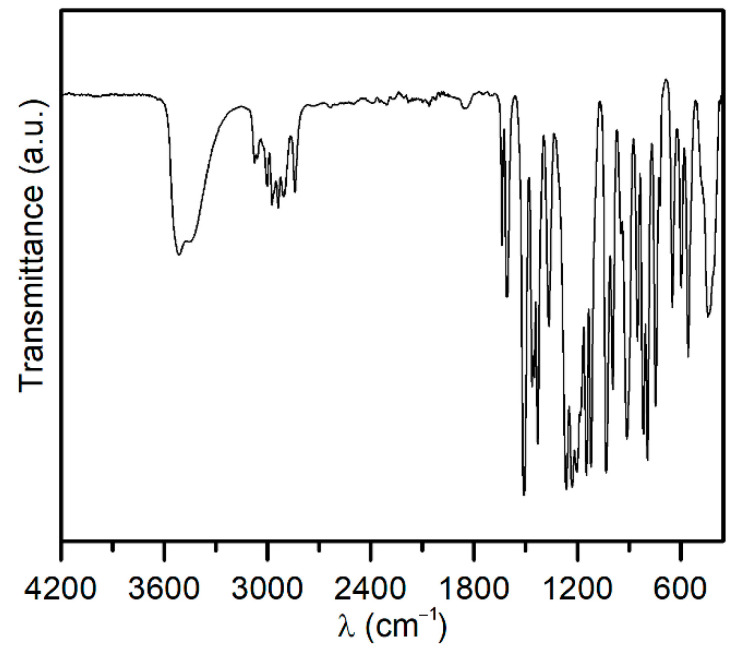
FTIR spectrum of eugenol.

**Figure 3 materials-15-04824-f003:**
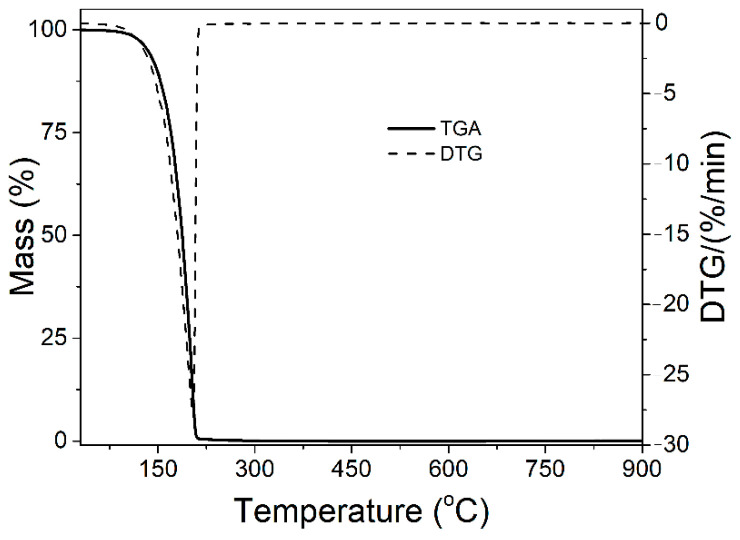
TGA and DTG plots of eugenol.

**Figure 4 materials-15-04824-f004:**
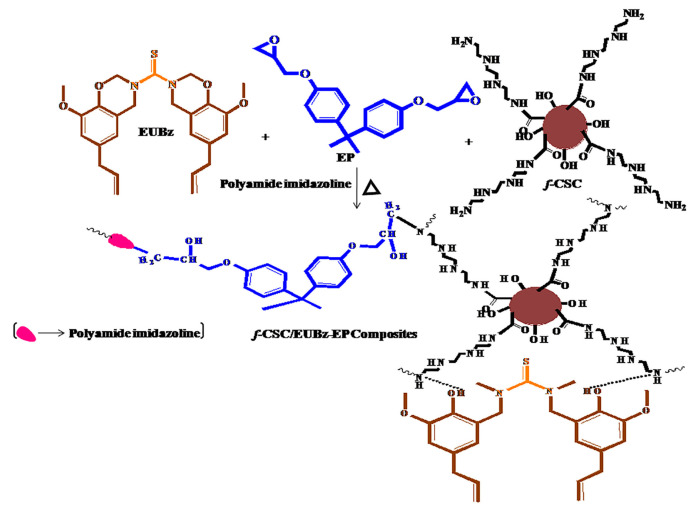
Scheme of preparation of epoxy composites with functional amine carbon from cashew nut shells using the eugenol–benzoxazine monomer (EBUz) based on thiourea (Reprinted with permission from *Polym. Compos* 2020, 41, 5, 1950–1961. Copyright 2020 John Wiley and Sons) [64].

**Figure 5 materials-15-04824-f005:**
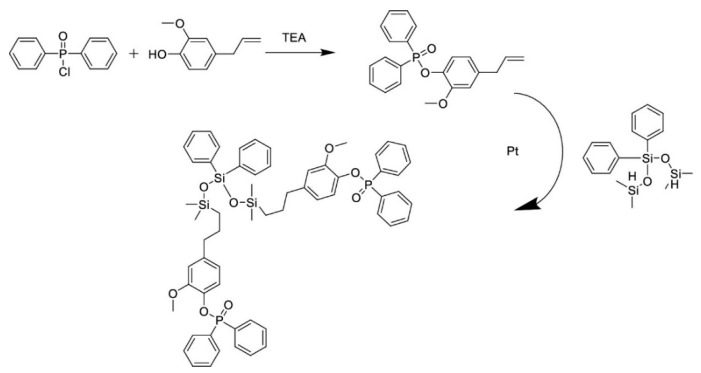
Scheme of synthesis of EGN-Si/P (Reprinted with permission from ACS Appl. Polym. Mater. 2022, 4, 3, 1794–1804. Copyright 2022 American Chemical Society) [69].

**Figure 6 materials-15-04824-f006:**
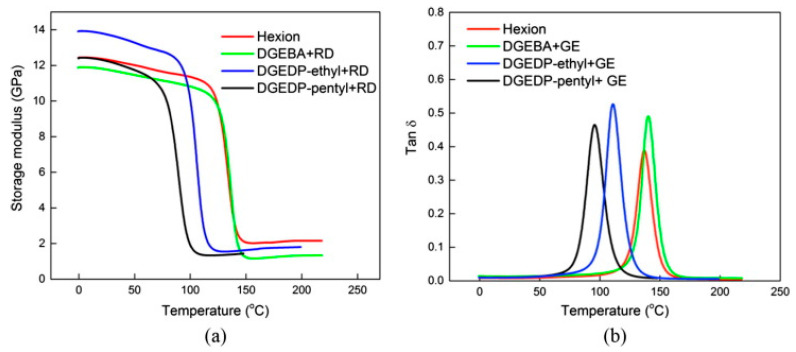
(**a**) Storage modulus and (**b**) Tan (δ) as a function of temperature for Hexion, DGEBA/15 wt% GE, and DGEDP/15 wt% GE epoxy/glass fiber composites. (Reprinted with permission from Compos. Part A Appl. Sci. Manuf. **2017**, *100*, 269–274 Copyright 2017 Elsevier) [70].

**Figure 7 materials-15-04824-f007:**
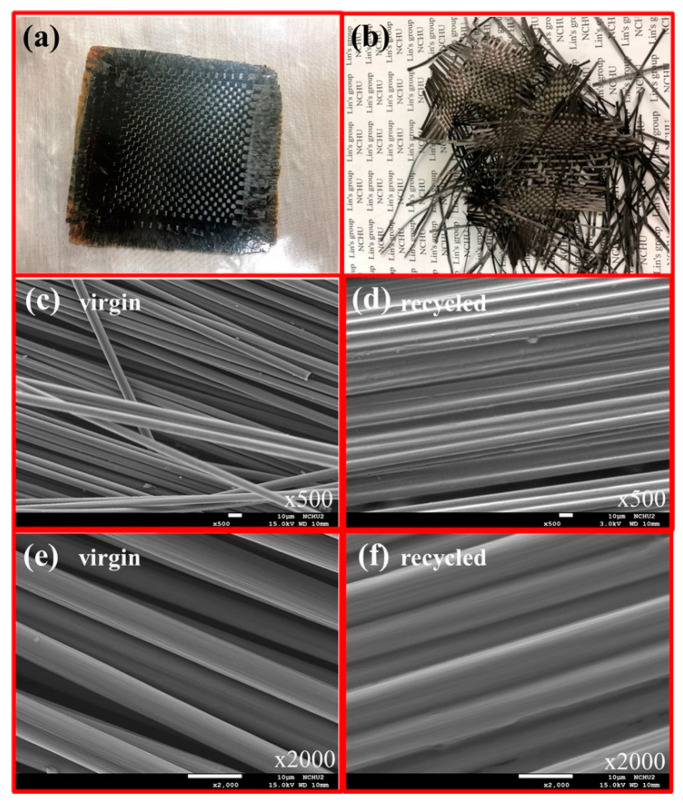
(**a**) Picture of the III/carbon fiber composite. (**b**) Picture of the recycled carbon fiber. (**c**,**e**) SEM pictures of the virgin carbon fiber. (**d**,**f**) SEM pictures of the recycled carbon fiber. (Reprinted with permission from ACS Sustain. Chem. Eng. 2021, 9, 5304–5314. Copyright 2022 American Chemical Society) [67].

**Figure 8 materials-15-04824-f008:**
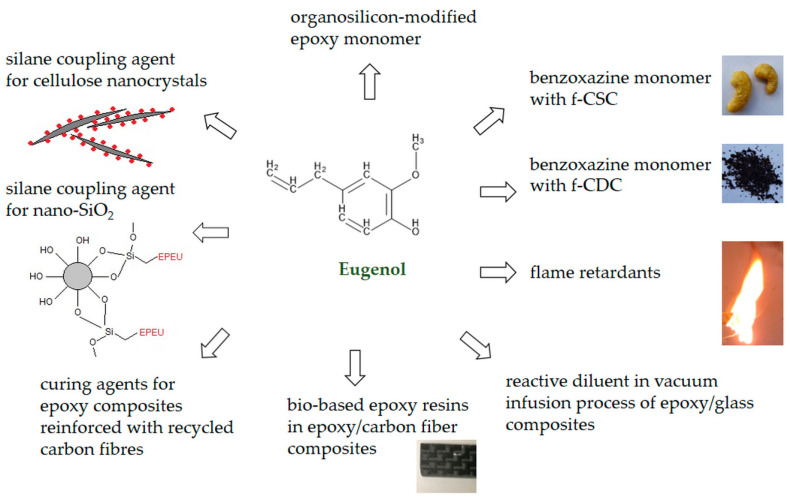
The prospect of using eugenol derivatives in epoxy composites.

**Table 1 materials-15-04824-t001:** Eugenol derivatives used in epoxy composites materials.

Name	Type	Synthesis Method	Ref.
eugenol based silicone coupling agent	coupling agent	hydrosilylation between triethoxysilane and 3-(4-allyl-2-methoxyphenoxy) propylene 1,2-oxide	[62,63]
eugenol–benzoxazine monomer (EBUz)	monomer	thiourea reaction	[64]
hetero structured benzoxazine monomer (HSBBz)	monomer	diphenyldiaminomethane reaction	[65]
organosilicon-modified epoxy monomer	monomer	melting polymerization of the epoxy group and amidogen with the disulfide bond	[66]
eugenol-based epoxy resins	monomer	- triethylamine and dichloromethane reaction, - 1,3,5-Benzenetricarbonyl trichloride reaction - meta-chloroperoxybenzoic acid reaction	[67]
diepoxy-isoeugenol phenyl phosphate (DEpiEPP)	flame retardant	- condensation of isoeugenol (iEu) and phenyldichlorophosphate in toluene with triethylamine, - epoxidation by oxone	[68]
eugenol-based flame retardant containing phosphorus (P) and silicon (Si)	flame retardant	- Williamson etherification reaction of eugenol and triethylamine in tetrahydrofuran i diphenylphosphinyl chloride, - silane reaction	[69]
eugenol monoglycidyl ether (GE)	diluent	glycidylation reaction	[70]
diamine-allyl-eugenol (DAAE)	curing agent	- Williamson reaction, thiol−ene radical reaction	[71,72]

**Table 2 materials-15-04824-t002:** Characteristics of epoxy composites with eugenol derivatives.

Matrix/ Curing Agent	Type of Filler	Eugenol Derivative	Filler wt. %	Producing Method	Improved Properties	Ref.
bio-based epoxy/ TETA	CNC	eugenol-based silane coupling agent (EBSCA)	1%, 3%, 5%	sonication, mixing, casting	tensile strength, storage modulus, degradability	[62]
epoxy (DGEBA)/ (IPDA)	SiO_2_	eugenol epoxy silane (EUPCP),	1%, 2%, 4%	sonication, mixing, casting	storage modulus, Tg temperature, thermal stability, flexural strength, impact strength	[63]
epoxy (DGEBA)/ polyamide imidazoline	f-CSC	eugenol–benzoxazine monomer (EUBz)	1%, 3%, 5%	mixing, casting	thermal stability, char yield, Tg temperature, anticorrosion futures	[64]
epoxy (DGEBA)/ polyamide imidazoline	*f*-CDC	hetero structured benzoxazine monomer (HSBBz)	1%, 3%, 5%	mixing, casting	thermal stability, char yield, Tg temperature, anticorrosion futures antibacterial properties	[65]
epoxy(DGEBA)/ CA bio-glycidyl ether of epoxyisougenol (GEEpiE)/CA	flame retardant	diepoxy-isoeugenol phenyl phosphate (DEpiEPP)	1%, 2%, 3%, 4.3% of P	mixing, casting	flame retardancy char yield,	[68]
epoxy (DGEBA)/ DDM	flame retardant	a eugenol-based flame retardant with phosphorus (P) and silicon (Si) groups (EGN-Si/P)	0.5% Si/0.37% P%; 1% Si/0.74% P; 2% Si/1.48% P	mixing, casting	flame retardancy. limiting oxygen index, impact strength	[69]
biobase diphenolate diglycidyl ethers (DGEDP)/ (IPDA)	E-glass fibre mats	eugenol monoglycidyl ether (GE) as a diluent	12 layers of glass mats,	vacuum infusion	viscosity reduction, gel time extension, storage modulus, flexural modulus, flexural strength,	[70]
tris(2-methoxy-4-(oxiran-2-ylmethyl)phenyl) benzene-1,3,5-tricarboxylate/ DMAP	carbon fibre	eugenol-based epoxy resins	6 layers of carbon fabric	mixing, immersing, pressing	degradability, rigidity, thermal stability,	[67]
resorcinol diglycidyl ether (RE)/(DAAE)	short carbon fibre	diamine-allyl-eugenol (DAAE) as curing agent	fiber volume content 27–32%	vacuum assisted resin infusion	tensile strength, bending strength thermal resistance,	[72]

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
