# Peer review of "Characteristics and Application of Eugenol in the Production of Epoxy and Thermosetting Resin Composites: A Review"

_materials, 2022, doi:10.3390/ma15144824_

Round 1
Reviewer 1 Report
Manuscript title- Euglenol as a bio component of epoxy and thermoset resin composites: a review. However, there are many lacks of presence in the manuscript. After major revision, the paper can be accepted for publication. Some specific comments are given below which require addressing properly.
1. The abstract is not presented in the proper way; this part should be rewritten according to the review article form. Moreover, the general discussion should be avoided from this section as seen in the beginning few lines.
2. Method: The selection process of cited articles is absent. It is essential to mention and discuss on what basis they are selected.
3. Major findings for all cited articles should be tabulated in the table and important observations of these are required to discuss briefly.
4. Preparation procedure of resin composite should be disused more clearly.
5. Some other properties like physical, morphological, and mechanical properties can be discussed.
6. Based on the findings of this review, a future direction is expected in the conclusion part.
Author Response
Dear Reviewer 1,
Thank you for your review, which will improve the quality of our article. The manuscript has been revised as recommended and all changes to the text are marked in red. The article has been supplemented with detailed information on the analysed composites. Two new tables with a comparison of the described materials have been added. Moreover, the English language was checked by a native speaker.
- The abstract is not presented in the proper way; this part should be rewritten according to the review article form. Moreover, the general discussion should be avoided from this section as seen in the beginning few lines.
Answer: The abstract has been revised as recommended
- Method: The selection process of cited articles is absent. It is essential to mention and discuss on what basis they are selected.
Answer: The process of selecting the cited articles is described in the text i.e.
“However, at present, a limited number of articles are available describing the preparation methods and features of thermosetting composites that were made by incorporating eugenol as a component or modifier”
“Articles describing epoxy composites in which eugenol was used to modify the polymer matrix or filler and as a modifier or hardener were selected for the review.”
“From among thermosetting composites, those whose polymer matrices were modified with eugenol derivatives were selected for review.”
- Major findings for all cited articles should be tabulated in the table and important observations of these are required to discuss briefly.
Answer: The recommended table has been prepared and references to it have been made in the text (table 2)
- Preparation procedure of resin composite should be disused more clearly.
Answer: In accordance with the recommendations of the preparation procedure of resin composite, it has been described in more detail for each material.
- Some other properties like physical, morphological, and mechanical properties can be discussed.
Answer: In accordance with the recommendations, a description of other properties that have been tested for the given composites has been introduced into the text
- Based on the findings of this review, a future direction is expected in the conclusion part.
Answer: The future direction of eugenol in composites is described in the conclusions.
Reviewer 2 Report
Environment friendly materials are needed to achieve sustainable society. The submitted review deals with eugenol as a bio component of epoxy resin. The contents is well written and it's acceptable after small corrections as follow.
1. L1: Englenol will be Engenol.
2. For example L168: "2" of SiO2 will be subscript.
3. The meaning of the short words are not clear. For example L144: DDS, L147: LOI.
Author Response
Dear Reviewer 2,
Thank you for your review, which will improve the quality of our article. The manuscript has been revised as recommended and all changes to the text are marked in red. Moreover, the English language was checked by a native speaker. The article has been supplemented with detailed information on the analysed composites. Two new tables with a comparison of the described materials have been added. Moreover, the English language was checked by a native speaker.
Reviewer 3 Report
The authors are addressing an important area of research on thermosetting bio-resins derived from eugenol for use as matrix and auxiliary agents in composites. However, the information is presented merely in the form of sporadic statements and/or specific data from the literature, giving little to be learned other than an exposure to some useful references. “A review should be a discussion of the data reported in the literature presented as a concise synthesis of the knowledge accrued from the work done in the area addressed”. This is clearly missing in the submission under consideration. One gets the impression that the authors are just reproducing statements and data from cited papers without digesting and scrutinize the information. For instance, in lines 237-239 are presented impact strength data in MPa units, without any indication of the implications and tests used. It is well known that impact strength denotes toughness which is usually expressed as fracture energy values in J/m2. Another example is the inconsistency of the information on the chemical nature of eugenol: There is a discrepancy between the structure in Figure 1 and Figure 8 and the description in the first sentence of the Discussion.
It is recommended that the paper should be re-written completely, and it is suggested that the topic is divided in 2 sections, respectively: 1) Synthesis and chemical characterization of eugenol derived resins and auxiliary components for composites. 2) Properties of composites based on eugenol derived resins and additives and coupling agents. This latter could be subdivided into Mechanical Properties and Fire Retardance. The introduction should present the prospects of eugenol as a potential raw chemical resource for producing resins and other agents or additives, from a point of view its reactivity, biocompatibility and commercial availability. This could be based on the scheme in Figure 8 with an elaborated discussion as premise for the work presented. In the text the authors should provide explanations and an interpretation of the data and information contained in Figures in sufficient details to make it unnecessary for the reader to have to consult the original source. Explanations in the form of new schemes and sketches to support the published materials should also be considered to help the reader. Finally, the title has to be improved.
Author Response
Dear Reviewer 3,
Thank you for your review, which will improve the quality of our article. The manuscript has been revised as recommended and all changes to the text are marked in red. The article has been supplemented with detailed information on the analysed composites. Two new tables with a comparison of the described materials have been added. Moreover, the English language was checked by a native speaker.
Comment 1: However, the information is presented merely in the form of sporadic statements and/or specific data from the literature, giving little to be learned other than an exposure to some useful references. “A review should be a discussion of the data reported in the literature presented as a concise synthesis of the knowledge accrued from the work done in the area addressed”.
Answer: According to the recommendations, the preparation and properties of the composites selected for review are described in more detail.
Comment 2: For instance, in lines 237-239 are presented impact strength data in MPa units, without any indication of the implications and tests used. It is well known that impact strength denotes toughness which is usually expressed as fracture energy values in J/m2.
Answer: This error has been corrected, the impact strength value has been given in accordance with the applicable unit, ie. J/m2.
Comment 3: There is a discrepancy between the structure in Figure 1 and Figure 8 and the description in the first sentence of the Discussion.
Answer: This error was corrected by the name of eugenol given in accordance with the preferred IUPAC name 2-Methoxy-4- (prop-2-en-1-yl) phenol.
Comment 4: It is recommended that the paper should be re-written completely, and it is suggested that the topic is divided in 2 sections, respectively: 1) Synthesis and chemical characterization of eugenol derived resins and auxiliary components for composites. 2) Properties of composites based on eugenol derived resins and additives and coupling agents. This latter could be subdivided into Mechanical Properties and Fire Retardance.
Answer: The manuscript was re-written. Details on the preparation and synthesis of composites of all eugenol derivatives have been added. Information on the synthesis of these compounds has been added to each description of composites. Additionally, Table 1 presenting methods of obtaining eugenol derivatives as bioadditives to composites was added, and a list of the properties of the composites is presented in Table 2.
Comment 5: The introduction should present the prospects of eugenol as a potential raw chemical resource for producing resins and other agents or additives, from a point of view its reactivity, biocompatibility and commercial availability. This could be based on the scheme in Figure 8 with an elaborated discussion as premise for the work presented.
Answer: The prospects of eugenol as a potential raw chemical resource for producing resins and other agents or additives, from a point of view its reactivity, biocompatibility and commercial availability are described in introduction.
Comment 6: In the text the authors should provide explanations and an interpretation of the data and information contained in Figures in sufficient details to make it unnecessary for the reader to have to consult the original source.
Answer: A detailed description of the figures has been introduced into the text. The title was also improved.
Comment 7:. Explanations in the form of new schemes and sketches to support the published materials should also be considered to help the reader. Finally, the title has to be improved.
Answer: In order to facilitate the comparison of the described epoxy composites, Table 1 and 2 with their comparisons has been added.
Round 2
Reviewer 1 Report
The paper is revised properly. Now it can be accepted for publication.
Author Response
Dear Reviewer 1,
Thank you very much for your reviews of our article. In addition, the article has been re-checked to correct the English language.
Reviewer 3 Report
Authors have made substantial improvements to both contents and presentation. The paper is now suitable for publication and will make a significant contribution to the literature.
Only minute revisions are required, e.g. 1) Add the word “resin” after epoxy and thermosetting in the Title, Abstract and Introduction. 2) Line 328, replace the word “retardant “ with retardancy.
.
Author Response
Dear Reviewer 3,
Thank you very much for your reviews of our article. Recommended corrections have been made and marked in blue in the text. In addition, the article has been re-checked to correct the English language.